# The Association of Pain with Incident Falls in People with Chronic Obstructive Pulmonary Disease: Evidence from the English Longitudinal Study of Ageing

**DOI:** 10.3390/ijerph20136236

**Published:** 2023-06-27

**Authors:** Kirsti J. Loughran, Daniel Tough, Cormac G. Ryan, Shaun Wellburn, Denis Martin, John Dixon, Samantha L. Harrison

**Affiliations:** Centre for Rehabilitation, School of Health & Life Science, Teesside University, Middlesbrough TS1 3BX, UK

**Keywords:** falls, pain, chronic obstructive pulmonary disease

## Abstract

People with chronic obstructive pulmonary disease (COPD) have a higher prevalence of pain and a greater risk of falls than their healthy peers. As pain has been associated with an increased risk of falls in older adults, this study investigated the association between pain and falls in people with COPD compared to healthy controls. Data from the English Longitudinal Study of Ageing were used to establish an association between pain and falls when modelled with a generalised ordinal logistic regression and adjusted for sex, age, wealth, and education (complete case analysis only; n = 806 COPD, n = 3898 healthy controls). The odds were then converted to the predicted probabilities of falling. The predicted probability of falling for people with COPD was greater across all pain categories than for healthy controls; for COPD with (predicted probability % [95%CI]), no pain was 20% [17 to 25], with mild pain was 28% [18 to 38], with moderate pain was 28% [22 to 34] with severe pain was 39% [30 to 47] and for healthy controls with no pain was 17% [16 to 18], mild pain 22% [18 to 27], moderate pain 25% [20 to 29] and severe pain 27% [20 to 35]. The probability of falling increased across pain categories in individuals with COPD, with the most severe pain category at a nearly 40% probability of falling, indicating a potential interaction between COPD and pain.

## 1. Introduction

Chronic Obstructive Pulmonary Disease (COPD) is a condition that causes airflow limitations in the lungs as a result of exposure to toxic particles and gases such as household and outdoor air pollution and tobacco smoke. COPD is characterised by respiratory symptoms such as dyspnoea, a cough, sputum production, and wheezing with reduced exercise capacity and activity limitation [1]. COPD also has a systemic effect, and its comorbidities are common; between 51% to 90% of people with COPD are estimated to have more than one comorbid condition, and up to 97% in those with GOLD stage 4 COPD [2,3,4,5]. Furthermore, frailty is common, with 23% of people with COPD being classified as frail and 56% as prefrail [6,7,8]. Falling is a prominent feature of the frailty syndrome, and therefore, it is unsurprising that people with COPD are four times more likely to fall than healthy adults without COPD. Falls have a significant economic and personal impact costing the National Health Service (NHS) in the United Kingdom over £2bn per year and the loss of 17 million disability-adjusted life years (DALYs) worldwide [9,10,11]. People with COPD who fall have reduced physical activity levels, a greater decline in their health-related quality of life, and increased mortality risk [12,13,14,15]. Pain is another extrapulmonary feature that is common in people with COPD; in fact, there is a 66% pooled prevalence of pain in people with COPD [16]. Pain in people with COPD is often severe, musculoskeletal in nature, and is often located around the thoracic region [16,17,18,19]. 

Despite the high prevalence of falls and pain in the COPD population, and the evidence suggesting that community-dwelling older adults with musculoskeletal pain are at a greater risk of falls, no study has investigated a potential association between pain prevalence in people with COPD and falls [20,21]. This is important because, currently, studies investigating the mechanisms of falls in risk and balance impairment and trials of interventions aiming to address the risks of falls with balance training or other interventions to improve balance, often in people with COPD, often exclude people with common musculoskeletal conditions or musculoskeletal pain [22,23]. This means that the impact of pain on fall risk and balance is unclear, and it is not known if such interventions are beneficial in reducing falls in those who may be more at risk.

COPD prevalence, musculoskeletal pain, frailty, and falls all increase with age (although musculoskeletal pain prevalence then reduces after the ninth decade), and with an ageing population, understanding how these factors interact is important in developing effective management and treatment strategies for falling [24,25,26]. Whilst it is not entirely clear why pain and falls are associated in older adults, it has been suggested that pain may interfere with the normal neuromusculoskeletal control of balance and gait. A meta-analysis of studies investigating balance in people with COPD highlighted a clinically significant impairment in their balance which was greater than in healthy controls; however, no studies in this review have investigated the impact of pain on either falls or balance in people with COPD [27]. 

The aims of this study were to (1) investigate the association between pain and incident falls in older adults with COPD, compared to healthy peers; (2) examine the impact of pain severity on the incident of a fall in older adults with COPD; and (3) explore the association between pain and incident falls requiring treatment (fall severity) in older adults with COPD and pain.

## 2. Methods

### 2.1. Study Design

Data from the English Longitudinal Study of Ageing (ELSA) were used to conduct a secondary analysis. The incidence of one or more falls in people with COPD was compared to that of people with COPD and no pain and to healthy controls over a follow-up period of two years from their baseline entry into the ELSA study. Odds ratios (OR) and predicted probabilities were reported to indicate the risk of falling.

ELSA is a large longitudinal panel study that is an extension of the Health Survey for England (HSE) and is designed to investigate aspects of ageing and older people in England [28,29,30]. ELSA began in 2002, collecting data on older people’s physical and mental health and well-being, as well as financial and social circumstances. Data were collected at two yearly intervals (‘waves’) via computer-assisted personal interviews and self-completed questionnaires. Visits by nurses were undertaken every four years to collect objective measures (e.g., blood samples and anthropometrics). ELSA refreshed the samples from HSE regularly to ensure younger age ranges were represented as the original sample aged; thus, participants joined the study at different waves. ELSA received ethical clearance from the London Multi-Centre Research Ethics Committee, including informed consent for sharing anonymised data via the UK data service [30]. 

### 2.2. Sample Selection

Participants were included in this data analysis if they were part of the ELSA cohort and were at least 58 years at the baseline assessment (because the question used for the falls variable two years later was only asked of those aged 60 years and above). Data were collected from waves 1–7 of the ELSA data (available at the time the data were downloaded in 2018), and variables of interest were extracted and cleaned. People with COPD were identified by creating a variable based on answering yes to the question “have you ever been diagnosed with…” and answering yes to the option of “… a chronic lung condition, e.g., Emphysema or chronic bronchitis?” as an answer. Asthma was a separate option that was available to answer this question, and therefore, people with asthma were not included in the COPD variable. Due to a far greater prevalence of COPD compared to other remaining lung conditions in older adults, most of this sample could be identified as having COPD. Smoking was also used as a cross-reference for COPD by selecting those who answered yes to the question “have you ever smoked?” in addition to a diagnosis of COPD because their lung function values were not available at this baseline, and this is a key exposure to the development of COPD in England. A control variable was created to include those who answered no to questions identifying any other chronic irreversible conditions with significant disabling symptoms that might affect their physical function; these included asthma, arthritis, osteoporosis, cancer, Parkinson’s Disease, emotional, nervous, or psychiatric problems, Alzheimer’s Disease or dementia, congestive heart failure, and stroke.

### 2.3. Exposure and Outcomes

A variable for the primary exposure of pain was created as a four-point scale of no pain, mild pain, moderate pain, or severe pain, based on answers to the questions “are you often troubled by pain?” and “how bad is the pain most of the time? Is it… mild, moderate or severe?”. A binary variable for the outcome of interest (falls) was created to include falls that occurred within a two-year period from the baseline wave of entry to ELSA. It was created based on the answer to the question, “have you fallen in the last two years?”. An ordinal variable of no falls, falls not requiring treatment, and falls requiring ‘medical’ treatment was also created using the questions “have you fallen in the last two years?” and “in that fall/any of the falls did you injure yourself seriously enough to need medical treatment?” to establish the severity of falls.

### 2.4. Covariates

Key covariates and confounders were identified using a direct acyclic graph (DAG) based on the current literature to model a fall causal pathway (Figure 1) [31,32]. A DAG is a visual representation of causal pathways using assumptions based on the available evidence where arrows indicate single or bidirectional associations that evidence suggests may have a causal link to each other. The DAG was used to aid the identification and selection of covariates for statistical modelling in combination with what data were available from the dataset. The covariates used in the modelling were age, sex, and education (highest educational qualification obtained from the categories of lower [no qualifications], intermediate [qualifications below college degree], or higher [college degree or above]) and wealth (a variable from the ELSA dataset of the baseline’s total net financial wealth calculated from financial and physical assets, business wealth, debt, primary housing wealth and mortgage debt, equity release, and home reversion plans, lifetime receipt of inheritance and gifts and life insurance) as representatives of socioeconomic status. Body Mass Index (BMI), pack-years (1 pack year = 20 cigarettes smoked every day for 1 year), and the number of comorbidities were considered but not included as multiple covariates as they could reduce the reliability of statistical testing and, in the case of BMI data, could have been recorded in a nurse wave at a four-yearly interval and thus was not a true reflection of baseline the BMI to be included in a causal pathway (i.e., BMI could have been recorded after a fall in the two years follow up period).

### 2.5. Data Analysis

The association between the primary exposure (pain) and the outcome (falls) was modelled using a generalised ordinal logistic regression, which was adjusted for sex, age, wealth, and education using a complete case analysis only (n = 9914) with Stata/IC 14.2 (StataCorp. 2015, Stata Statistical Software: Release 14, StataCorp LP, College Station, TX, USA). Odds ratios (ORs), with 95% confidence intervals, were calculated for its association between exposure to pain at the baseline (a first wave that the participant entered the ELSA study) and incident falls in the following two-year period (data collected at the next wave of data collection following entry to the study). For ease of interpretation, ORs were converted to predicted probabilities of falling to provide an indicator of absolute risk in percentage terms, and data for people with COPD and healthy controls were then extracted to limit the potential of sparse data bias affecting the analysis due to frequent missing data across several variables [33]. The probabilities were derived at the mean value of continuous covariates and factor variables treated as balanced. This analysis was repeated with the falls either requiring treatment or no treatment variable using the same model; data only for people with COPD were extracted to give the predicted probabilities of fall severity in people with COPD. The odds ratios for this analysis are not available as the variable for fall severity has three levels. The predicted probabilities were derived in the same manner as the first analysis. 

## 3. Results

A total of n = 18,572 people aged 58 and over were included in the initial database. After the removal of cases with missing data, n = 9914 were available for a complete case analysis; n = 2072 had missing data on included covariates (age, sex, wealth, and/or education), and n = 6586 were missing fall variable data. Once the data for groups were extracted from the complete case data, there were n = 806 people with COPD and n = 3898 healthy controls (Figure 2). Once these missing data on the severity of falls were excluded, the second analysis featured n = 585 people with COPD. 

Both groups were similar in terms of their age, sex, and BMI (see Table 1). These differences were seen in the percentages of people in each group experiencing pain: 56.1% of the COPD group experienced pain, compared to 22.4% in the healthy control group. The COPD group had a higher percentage of people reporting falls (33%) than the healthy control group (19.5%). People with COPD also had a lower wealth (£28K COPD vs. £65K healthy controls) and education status (median no qualifications COPD and median intermediate healthy controls).

The odds ratios revealed a linear relationship between the pain severity category and incident falls for both people with COPD and the healthy controls (Table 2). People with any severity of pain in both groups had a greater risk of falling than those without pain, which increased with pain severity (OR 1.47 COPD and 1.40 healthy controls for mild pain, OR 1.68 COPD and 1.61 healthy controls for moderate pain, and 2.39 for COPD and 1.84 for healthy controls for severe pain). The predicted probabilities can be seen in Table 3.

In people with COPD, the predicted probabilities of incident falls not requiring treatment were 15% for those with no pain (95%CI 12 to 19), 13% for those with mild pain (95%CI 6 to 20), 21% for those with moderate pain (95%CI 16 to 27) and 29% for those with severe pain (95%CI 21 to 36) (Table 4). However, no such relationship was seen in people with COPD for the predicted probability of a fall requiring treatment, with low probabilities across all the pain severity categories (Table 4). People with COPD and no pain had a 6% predicted probability of a fall that required treatment (95%CI 3 to 8), compared to 15% for those with mild pain (95%CI 7 to 22), 7% for those with moderate pain (95%CI 4 to 10), and 10% for those with severe pain (95% CI 5 to 14).

## 4. Discussion

This is the first study to investigate an association between pain and incident falls in people with COPD. The predicted probability of falling for people with COPD and pain was greater than for both people with COPD and no pain, and for healthy controls with pain. As the severity of pain increased for both groups, so did the predicted probability of falling, but this was greater for those with COPD across all pain severities. Those with COPD and severe pain were at an almost 40% predicted probability of falling, indicating the potential interaction between COPD and severe pain. An incremental increase in the predicted probability of falling for those with COPD who did not need medical treatment (less severe falls) was also noted. However, although those with COPD and pain those who did require treatment had a higher predicted probability of falling than those with COPD and no pain, no such incremental increase was seen.

Despite being similar in terms of age, sex, and BMI, people with COPD had a lower socioeconomic status (in terms of education and wealth) compared to healthy controls. COPD has been associated with lower socioeconomic status, and therefore, this sample appeared to be representative of the COPD population [34]. The COPD group had a higher percentage of those who experienced pain compared to the healthy control group, which is in line with the pain prevalence cited in previous studies when investigating pain in people with COPD: 56.1% in this study vs. 66% in Lee et al., 2015 [16]. The COPD group also had a higher percentage of fallers as per the previous literature, which compared falls in people with COPD vs. healthy controls (33% in this study vs. 15% in healthy older adults) [35,36,37]. 

Pain, especially musculoskeletal pain, has been reported to increase the risk of falling in older adults [38,39,40]. People with COPD have a high prevalence of pain which is mostly musculoskeletal in nature and may explain the greater predicted probability of falling for people with COPD and pain compared to healthy controls with pain. Those with COPD and severe pain had the highest predicted probability of falls at nearly 40%, indicating the possible interaction between having COPD and severe pain. Evidence from older adult populations also suggests that severe pain, which is often multi-site, can also be associated with an increased fall risk [41,42]. People with COPD often have more pain-associated comorbidities and, thus, may experience more severe and/or multi-site pain; in addition to the common associates of pain such as Osteoarthritis (OA) in weight-bearing joints, people with COPD often reported pain in the thoracic region [17,43]. Severe pain may also increase the risk of falls by causing greater pain interference [44,45]. Psychological factors such as fear of falling and falling efficacy are associated with pain and increased risk of falls; fear of falling has been reported to be higher in people with COPD than in any matched controls [21,46].

The mechanisms underlying an observed increase in the risk of falling in people with COPD are unclear, but there are areas of overlap between COPD and pain that may interact. Painful comorbidities such as OA are common for people with COPD, and evidence suggests that pain, rather than the structural changes caused by comorbidity, may have an inhibitory effect on the neuromuscular control of balance and activity in relation to falls [47]. Lower physical activity levels are also common in people with COPD and those with pain; pain may have a cyclical effect by further reducing physical activity and increasing pain [48,49]. Both COPD and pain may impact balance, which is a known risk factor for falls, and deficits in balance have been shown in people with COPD and in chronic pain populations [27]. However, it is unclear if pain is also associated with balance impairments, as seen in people with COPD [27]. This is important because both balance and pain are modifiable factors where appropriate interventions and improvements in both could reduce fall risk [50]. The intervention development for fall management in people with COPD should consider how interventions can be delivered in the presence of pain rather than excluding those with pain from the study cohorts. There is a growing body of evidence to support the benefits of pulmonary rehabilitation (PR) with balance training on improving the balance outcomes in people with COPD. Yet, pulmonary rehabilitation has been described as a short-term aggravator of pain, and pain has also been cited as a reason for people dropping out of pulmonary rehabilitation programmes [51]. Pain is seldom discussed during the initial assessment for PR; however, this might be an opportune time to discuss individual concerns about pain [52].

The lack of an incremental increase in the risk of falls in those who require treatment could have been reflected by the smaller sample sizes and wider confidence intervals of this sub-sample and the ambiguity of the variable—with no definition of the type of ‘medical’ treatment provided in the ELSA literature. People with COPD had greater odds (1.18) of an in-hospital fall, which is related to major injuries and falls that often precipitate transfers to care or mortality; thus, the most severe falls requiring treatment may not have remained in the sample if their data were not available [53,54,55]. Although proxy interviews with a friend or relative were offered in these scenarios, the stigma and recall bias of transferred information on the recording of retrospective falls may mean that the history of falls was not accurately recorded. In COPD, falls can predict mortality and survival after a fall-related fracture with COPD is poor, with a 60–70% higher risk of death following a hip fracture compared to those without COPD [15,56]. Those in the ELSA cohort who passed away would not have had follow-up data and, thus, would have been excluded from the analysis because end-of-life proxy interviews do not include fall data.

This study had a number of limitations. The use of secondary data limited the design of the studies. In this instance, the clinical outcome data taken at nurse wave visits, particularly lung function measurements and BMI data, could not be used. The nurse visits were optional and, thus, subject to missing data to a greater extent than other variables. The nurse visits could have been up to two years after the baseline measurements; therefore, BMI data were not reliable to use as an adjustment in a causal pathway. The falls variable was also a retrospective method of recording which has been highlighted as less accurate and subject to recall bias and stigma. The entry to this study was between 2002 and 2015, and incidence data were collected between 2003 to 2017 and thus may not fully reflect current trends. It should also be noted that these data were collected and analysed prior to the COVID-19 pandemic, and the impact of the pandemic on pain and falls is unknown. It was also not possible to identify all comorbidities beyond the common comorbidities identified by the main ELSA dataset; therefore, some of the healthy control group may still have had comorbidities that impacted their risk of falls. However, this study used a complete case analysis, with the extraction of groups of interest to allow for any sparse data bias, and large data allowed for large sample sizes for the first analysis, adding to the robustness of these findings. The use of a secondary data set also allowed for the consideration of a causal pathway considering exposure and a two-year follow-up period, which would otherwise have been costly and time-consuming. Using a direct acyclic diagram to inform the variable selection based on evidence is also a strength.

## 5. Conclusions

People with COPD and pain have an increased predicted probability of falling. Those with severe pain and COPD are at the highest risk of falling. There may be shared factors between COPD and severe pain that interact to increase the risk of falls, such as multi-site pain, pain interference, and psychological impacts. Further prospective research is required to determine if the pain is associated with falls in people with COPD, including the identification of those with pain for intervention to prevent the deleterious effects of falling.

## Figures and Tables

**Figure 1 ijerph-20-06236-f001:**
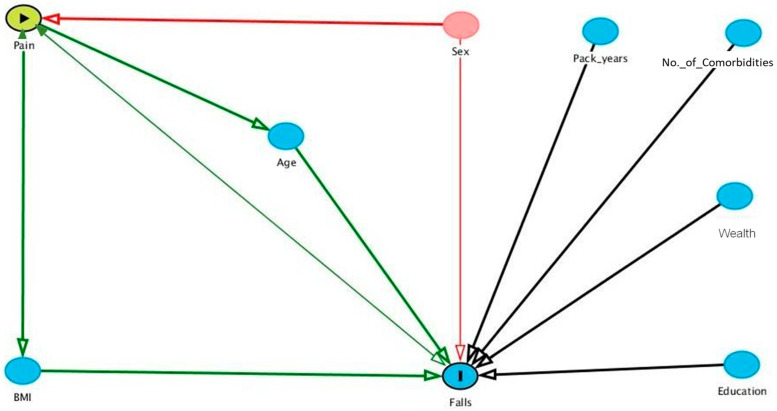
Direct acyclic graph of the potential causal pathway of falls in people with COPD. Green lines = causal impact; red line = moderator of exposure or outcome; black line = an association with outcome.

**Figure 2 ijerph-20-06236-f002:**
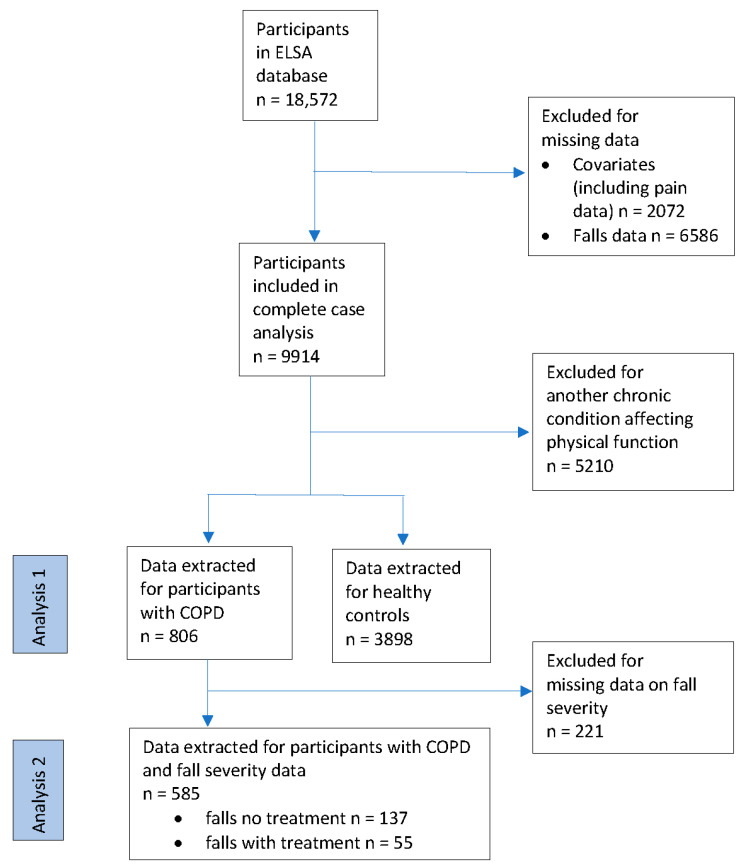
Flow diagram of the study participants.

**Table 1 ijerph-20-06236-t001:** Demographic data for the COPD and healthy control groups.

	COPD Group (n = 806)	Controls (n = 3898)
Mean age (SD)	69.6 (7.6)	68.2 (7.7)
% female	43.3%	44.3%
Mean BMI (SD)	27.9 (5.5)	27.5 (4.4)
Experience pain	56.1%	22.4%
% fallers	33%	19.5%
Wealth (£) (SD)	£28,477.62 (£115,349.00)	£65,097.28 (£198,067.50)
Education (median, IQR) 1 = No qualification 2 = Intermediate 3 = Higher	1 (1)	2 (2)

SD = standard deviation; IQR = Interquartile Range.

**Table 2 ijerph-20-06236-t002:** Risk of incident falls (OR) in those with COPD and pain vs. healthy controls, which were fully adjusted for the model.

	Mild Pain (95%CI)	Moderate Pain (95%CI)	Severe Pain (95%CI)
Falls COPD	1.47 (0.86 to 2.54) *p* = 0.161	1.68 (1.01 to 2.18) *p* = 0.046	2.39 (1.56 to 3.66) *p* = 0.000
Falls controls	1.40 (1.06 to 1.85) *p* = 0.018	1.61 (1.26 to 2.06) *p* < 0.000	1.84 (1.24 to 2.75) *p* < 0.003

*p* values indicate a comparison between that category and the reference value of no pain.

**Table 3 ijerph-20-06236-t003:** Predicted probabilities (%) of incident falls in those with COPD and pain vs. healthy controls by pain severity.

	No Pain (95%CI)	Mild Pain (95%CI)	Moderate Pain (95%CI)	Severe Pain (95%CI)
Falls COPD	20 (17 to 25)	28 (18 to 38)	28 (22 to 34)	39 (30 to 47)
Falls controls	17 (16 to 18)	22 (18 to 27)	25 (20 to 29)	27 (20 to 35)

**Table 4 ijerph-20-06236-t004:** Predicted probability (%) of incident falls requiring treatment or no treatment in people with COPD by pain severity.

	No Pain (95%CI)	Mild Pain (95%CI)	Moderate Pain (95%CI)	Severe Pain (95%CI)
Falls no treatment n = 137	15 (12 to 19)	13 (6 to 20)	21 (16 to 27)	29 (21 to 36)
Falls with treatment n = 55	6 (3 to 8)	15 (7 to 22)	7 (4 to 10)	10 (5 to 14)

*p* values indicate a comparison between that category and the reference value of no pain.

## Data Availability

The data presented in this study are available on application to the UK Data Service (https://ukdataservice.ac.uk/find-data/access-conditions/) (accessed on 16 June 2023). Data were accessed on 4 July 2017.

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
