# Peer review of "The Association of Pain with Incident Falls in People with Chronic Obstructive Pulmonary Disease: Evidence from the English Longitudinal Study of Ageing"

_ijerph, 2023, doi:10.3390/ijerph20136236_

Round 1
Reviewer 1 Report
This is a relevant study investigating the association between pain and the incidence of falls in people with COPD compared to healthy peers. The study also examines the impact of pain severity on incident falls in people with COPD. I congratulate the authors for their effort in using data available in the ELSA cohort. The new information provided was based on robust sample size. The manuscript is well-written and structured. However, more information must be included, which may ease the readership's understanding. Please, find below a few major and minor concerns:
Major
It needs to be made clear how the physical function of controls was assessed in ELSA, if possible, and describe this in the Methods section. In the Results section, describing which chronic conditions were identified as affecting physical function would also be informative, as stated in Figure 2.
The statistical analysis for between-group comparison should be provided in Table 1.
In the Discussion, a comment about the potential effect of comorbidities in the fall prevalence recorded in the control group is relevant. Could it also underestimate the differences in fall prevalence observed between COPD and non-COPD? Also, the use of 'healthy' to characterize controls should be revised throughout the text, as in a great cohort like ELSA, other comorbidities are highly expected in this group.
Minor
The percentage symbol is missing for moderate pain in COPD in the Abstract.
A reference is missing in the second paragraph of the Introduction.
Please revise the title in Table 2.
Reviewer 2 Report
General: the passive voice language is a bit hard to read and follow. Eg in methods paragraph starting with line 88. I would prefer the active voice "we created a variable..."-- perhaps can discuss the preference for active vs passive voice with editors.
Line 11: can you say something more definitive/clearer about the known link between pain and falls and clinical implications? Increased pain leads to more falls? Would help to set up the knowns vs unknowns in the paper.
Line 20: can you also include a summary statement of the difference in falls/between COPD and healthy controls? I found myself reading the list of % and then trying to compare to see what the difference was, but it is hard to do.
Line 26: I don't think the description of COPD as only an inflammatory condition is correct. The GOLD report cited says that COPD has inflammatory and structural changes components (pages 15-16 of 2023 GOLD report). Also recommend citing most recent GOLD report, they come out yearly.
I think the introduction could also be strengthened by including age and/or geriatric conditions as a connecting discussion point between some of the things you are looking at (COPD prevalence increases with age, as does musculoskeletal pain, frailty, and falls) so all of these issues are particularly relevant in the context of an aging society (and especially important to discuss since you are using a dataset focused on aging)
Finally, is there any proposed mechanism for why people might have more falls if they have more pain? It would be useful to understand why these two things may be connected as we start the paper to give support to the clinical relevance of the findings (I believe this is in the discussion, perhaps one line in the intro would suffice)
Line 47: Missing references
Line 48: Would suggest instead of saying "people with COPD" would say "older adults with COPD" as your study group is age-limited by design
Line 57: should be "odds ratios" not "odds ratio's"
Line 61: should be "older people's" not "older peoples"
Line 77: can you please clarify your case definition for COPD? I am confused by the discussion involving asthma-- do you mean that people saying yes to "have you ever been diagnosed with a chronic lung condition" could mean they have asthma, and you excluded people who did not smoke?
Line 99: the DAG (direct acyclic graph) needs a bit more explanation (what it is, why was it created for this study)
Line 101: I am not familiar with the UK educational system (I'm American) so would be useful to explain how you defined education a bit more, as I don't know how to interpret this variable. eg I don't know what "no qualification" means
Line 112: can you be clearer about which wave you took falls info from and which you took pain from? same wave? you mentioned using waves 1-7-- was it cross sectional but just took participants from different waves?
Line 146: should be "odds ratios" not "odds ratio's"
Line 103: I am confused what wealth means and your definition didn't clear it up for me (baseline total net financial wealth). Does this mean money in the bank? Including all assets and subtracting all debts? Could consider creating a supplement to expand more on these variable definitions
Line 144: figure 2: can you include in the text what conditions you excluded participants for having had another condition that limits physical function?
Line 169: This summative statement is confusing and would benefit from the active voice
